Research

# Determining a reliably visible and inexpensive surface fiducial marker for use in MRI: a research study in a busy Australian Radiology Department

Maree T Izatt,[1] Deborah Lees,[1] Susan Mills,[2] Caroline A Grant,[1] J Paige Little[1]

¹Biomechanics and Spine Research Group, Queensland University of Technology, Brisbane, Queensland, Australia
²Mater Medical Imaging, Mater Misericordiae Brisbane Ltd, South Brisbane, Queensland, Australia

**Correspondence to**
Dr J Paige Little;
j2.little@qut.edu.au

## ABSTRACT

**Objectives** Single-use commercial surface fiducial markers are used in clinical imaging for a variety of applications. The current study sought to find a new, reliably visible, easily sourced and inexpensive fiducial marker alternative for use with MRI.

**Design** Five commonly requested MRI sequences were determined (three-dimensional (3D) T1-weighted, T1 coronal, 3D T2-weighted, T2 fat suppressed, proton density), to examine the visibility of 18 items (including a commercial fiducial marker).

**Setting** Clinical 3T MRI scanner in an Australian Tertiary Hospital and an Australian University Biomedical Engineering research group.

**Interventions** 18 marker alternatives were scanned using five common MRI sequences. Images were reformatted to obtain both an image through the mid-height of each marker and a maximum intensity z-projection image over the volume of the marker. Variations in marker intensity were profiled across each visible marker and a visibility rating defined.

**Main outcome measures** Outcome measures were based on quantitative assessment of a clear intensity contrast ratio between the marker and the adjacent tissue and a qualitative assessment of visibility via a 3-point scale.

**Results** The fish oil capsule, vitamin D capsule, paint ball pellet, soy sauce sushi tube and commercial markers were typically visible to a high quality on all the imaging sequences and demonstrated a clear differential in intensity contrast against the adjacent tissue. Other common items, such as plasticine 'play doh' and a soft 'Jelly baby' sweet, were surprise candidates, demonstrating high-quality visibility and intensity contrast for the 3D T1-weighted sequence.

**Conclusions** Depending on the basis for referral and MRI sequence chosen, four alternative fiducial markers were determined to be inexpensive, easily sourced and consistently visible. Of these, the vitamin D capsule provided an excellent balance between availability, size, cost, usability and quality of the visualised marker for all the commonly used MRI sequences analysed.

## Strengths and limitations of this study

► This manuscript is the first to test for easily sourced and economical items to find reliably visible surface fiducial marker options for common MRI sequences.

► Scanning the fiducial markers options on the thigh of a healthy adult female provides a superior assessment of marker visibility than analyses that use a saline phantom object.

► The study presented both quantitative and qualitative analyses of marker visibility, thus providing a more practical assessment of the ability of the radiographer/investigator to visualise the marker.

► Five commonly utilised MRI sequences were investigated to provide the most relevant alternative marker for the majority of clinical MRI but there may be other MRI sequences of interest that were not included in the current study.

► The markers tested in the current study were a limited sample of convenience and included only one commercial fiducial marker.

as distinct regions of high intensity to assist in pinpointing specific anatomical landmarks or pathologies on the acquired clinical images. Surface markers placed on a patient's skin can also provide a frame of reference for registration of medical images acquired using multiple imaging modalities, such as photography, computed tomography (CT) and magnetic resonance imaging (MRI) performed for concomitant pathologies involving multiple specialities.[1]

In radiotherapy, fiducial markers are increasingly being used to identify the tumour site, permitting image registration and assisting with guidance for treatment planning.[2–4] Oil-based surface markers have been shown to compare favourably to solid markers in terms of their contrast to noise ratio, resulting in excellent visibility.[4] Prostate marker studies also revealed that implanted titanium seeds left star/streak artefacts on CT imaging and could not be

## INTRODUCTION

Fiducial markers used in the context of clinical medicine may be implanted in the body or placed on the surface and are also utilised in research for a variety of applications. Placed in the field of view, they display

accurately localised on MRI due to negative contrast (black holes).[5–7] Fiducial markers also perform a valuable role in guiding anatomical identification in musculoskeletal studies, particularly kinematic gait analyses where the location of surface markers on both mathematical gait models and the accompanying three-dimensional (3D) MR images provides valuable subject-specific simulation parameters.[8]

Commercially available fiducial surface markers (ie, non-implantable) are available in a range of sizes and shapes for use in the wide variety of clinical imaging applications. They are self-adhesive and have a low, flat profile, making them comfortable for the patient. While these markers provide excellent utility, they present a substantial expense to radiology departments, with a review of the current markets showing prices ranging from $420 to $600 USD for a box of 100 (MR Spots and IZI multi-modality, excluding shipping and tax). Being a single-use product, they are a considerable expense to the running costs of imaging departments or research teams utilising MRI and requiring numerous markers.

A review of the literature to find materials that are visible in MRI revealed a paucity of publications. The majority of papers examining substance visibility were focused primarily on locating a variety of penetrating or ingested foreign bodies and just a few papers explored implanted marker options for tumour boundary markers prior to excision or radiotherapy planning.[3 5–7 9] Foreign bodies that had been located, ranged in material from fish and chicken bones, batteries, plastics, coins, metal and wood splinters[10–13] and it should be noted that some materials were missed even with the high sensitivity of modern MRI and CT.[14 15] Pattamapospong *et al*[16] published results regarding the visibility of a number of substances on X-ray, CT and MRI and demonstrated a 100% specificity but only 58% sensitivity for fresh wood, dry wood, glass, plastic and porcelain using MRI. Wax crayons have proven to be easily visible on both CT and MRI with high attenuation noted on CT and a signal void on both T1 and T2 MR images. Interestingly, it was noted that crayon colour influenced the degree of visibility. Different pigments used in crayons and in paints resulted in distinctive colours as well as MRI appearance.[17] Injectable facial fillers (hyaluronic acid, collagen and polyalkylimide–polyacrylamide hydrogels), silicone and calcium hydroxyapatite have also been well visualised on MRI but can be confused with malignant features, so are not considered good marker substances.[18] Metal is well visualised on many modalities[10] but due to its ferrous properties, is contra-indicated in the field of MRI. Metal, even if MR conditional, results in significant image attenuation and artefact. For these reasons, an iron tablet as a makeshift marker may be well visualised but the resulting artefact renders it unacceptable. Studies looking at the visibility of plastic, stone, glass and graphite revealed that both glass and plastic foreign bodies are not consistently visible on MRI or plain radiographs, despite showing up well on CT.[16 19]

A review of the physics of MRI suggested materials with high water or fat content were likely items that would be readily detectable. Fiducial markers should be easily identifiable and clearly visible on clinical images, allowing the target anatomy or volume to be clearly and completely visualised. Therefore, a critical objective of this study was to assess the validity of various 'everyday' items, which could be easily and economically sourced, and provide a reliably visible fiducial marker as an alternative to the comparatively expensive, single-use commercial fiducial markers. We hypothesised that an inexpensive, readily sourced, robust surface fiducial marker could be isolated, that consistently demonstrated at least the same level of visibility as a commercial fiducial marker, when viewed on the most commonly performed MRI sequences.

## METHODS

Seventeen everyday items as well as a commercial fiducial marker were selected for analysis, either from the literature or anecdotal reports (figure 1). The important considerations relating to the selection of a suitable surrogate for commercial fiducial markers were identified and are outlined in table 1. In addition to the commercial fiducial marker, two of these items had been used in our local hospital medical imaging department; the paint ball (PB) for MRI scans in a pilot research project, and the fish oil (FO) capsule which was in regular use to avoid using a commercial marker (CM) whenever possible. While these two non-CMs provided excellent results in terms of intensity contrast and visualisation on MR images, to date a definitive comparison of the image contrast provided by these items has not been undertaken and these alternatives did not meet all the desired requirements of the ideal alternative fiducial marker.

### Imaging details

The selection of items underwent MRI scanning in a clinical medical imaging department using a 3T Philips Achieva MRI scanner. Scanning parameters vary between different machine manufacturers and even between different scanner models from the same manufacturer. However, this particular scanner was selected as it services a busy hospital radiology department in a metropolitan city and, thus, provides a broad range of exploratory and treatment-based MRI scanning services. Further to this, the 3T magnet strength is a typical specification for scanners in tertiary care facilities in Australia and internationally.

Following the advice from the senior MRI radiographers and a survey of local surgeons, five different sequences were performed (table 2) to cover the breadth of MRI studies typically performed with a requirement for inclusion of surface markers. All tested markers were attached to the anterior thigh of a healthy female participant, aged 27 years, who had provided informed consent. A radiofrequency coil (eight-channel knee coil) was positioned beneath the thigh and the markers attached

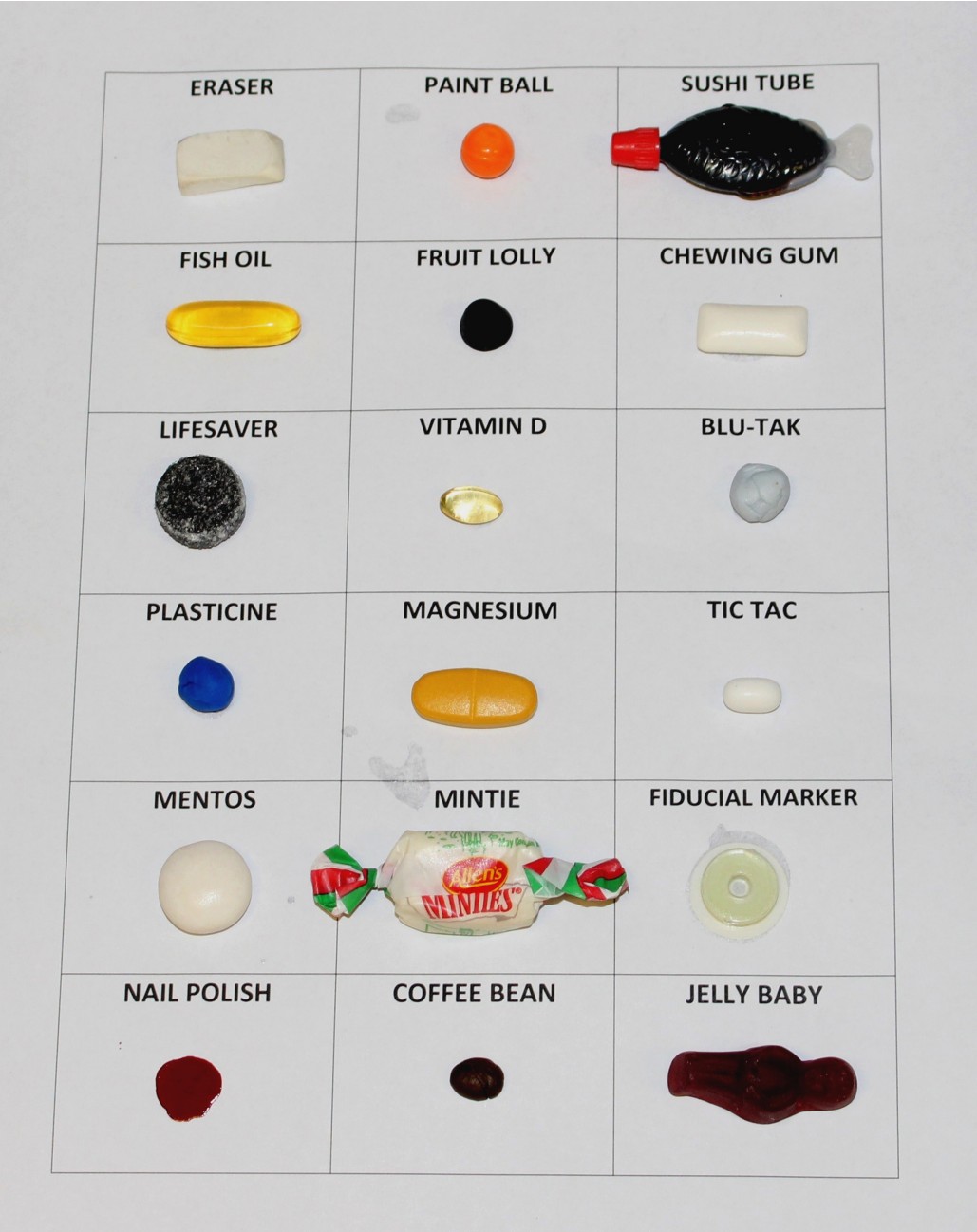

**Figure 1** 18 items investigated as surrogates for the commercially available fiducial markers—eraser, paint ball, sushi tube, fish oil capsule, fruit lolly, chewing gum, lifesaver, vitamin D, blu-tak, plasticine, magnesium tabsule, tic tac, mentos, mintie, commercial fiducial marker (IZI Multi-Modality, LifeHealthCare, Australia, included as a benchmark), nail polish, coffee bean, jelly baby sweet.

to the anterior thigh surface (figure 2). Due to physical size constraints, the marker selection was attached to the volunteer's leg in two separate acquisitions. In all, the MRI sequences detailed in table 2, the scanning parameters were constant for both acquisitions.

Images were saved in Digital Imaging and Communications in Medicine (DICOM) format and analysed using ImageJ (US National Institute of Health Open-source software, Maryland, USA; https://imagej.nih.gov/ij/). A similar process was utilised to assess the results from each of the five different imaging sequences (table 2) and involved first viewing the 3D stack as a segmented volume using the ImageJ Plugin, 'Volume Viewer 2.01' (K U Barthel, Internationale Medieninformatik, HTW Berlin, Germany) (figure 3).

In the first instance, a qualitative evaluation of the marker visibility was carried out to subjectively assess the relative ease with which each marker could be visualised and the marker edges demarcated. Viewing the data as a 3D volume permitted an initial assessment of which markers exhibited an intensity that was visible on MRI (figure 3). If the marker was visible, the visibility was rated on a 3-point Likert scale with 3 representing 'very clear', 2 representing 'edges visible but fuzzy' and

| Table 1 | Important considerations relating to the choice of a suitable fiducial marker for MRI |
|---|---|
| **Factors for consideration** | |
| Accessibility | Easily acquired, readily replaced and preferably locally sourced without high shipment costs or transit time |
| Use | Sufficiently robust to tolerate body weight if placed between the participant and scanner bed |
| | Suitably compliant or flat to ensure minimal discomfort to the patient if marker is in a load-bearing location (eg, posterior superior iliac spine for a supine MRI) |
| Biochemistry | Chemically inert, hypoallergenic |
| Economy | Low acquisition cost per marker given markers are single-use items |
| Medical physics | Sufficient hydrogen content to permit high intensity contrast to surrounding tissue |
| Fixation | Reliably affixed to the patient's external anatomy |

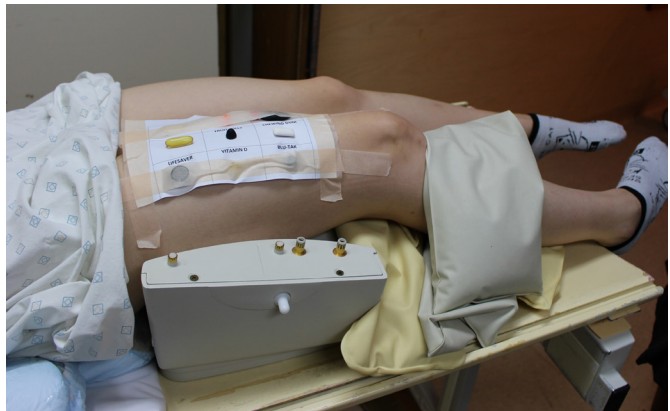

**Figure 2** Markers were rigidly attached to the anterior surface of the thigh and the upper leg positioned in a radiofrequency coil. These images show the first nine items investigated.

1 representing 'not easily visible'. If a visibility rating of 1 was observed, this low rating suggested that while the marker was discernibly visible on the MR images, the quality of visibility was poor and the marker would not be recommended for use in clinical scanning using that particular MRI sequence. Using this scale, a mean visibility rating across the five MRI sequences was calculated for each fiducial marker. While a mean visibility rating >0 indicated that the marker could be seen on at least one of the MRI sequences, a mean visibility rating greater than 2 was preferable as it indicated a consistently good quality visibility on all of the MRI sequences performed.

In all cases, MR images were reformatted and processed as 16-bit images, with an intensity range 0 to 65 536. A quantitative evaluation of the marker quality was carried out to assess the visibility, artefacts and distortion of marker boundaries created by each item on each MRI sequence. This evaluation was carried out using the steps outlined below and shown visually in figure 4.

1. Reformatted Image Stack: The DICOM stacks were reformatted to create a sequential stack of transverse reslice images, with consideration of the relevant pixel and slice spacing for each dataset (table 2).
2. Intensity Profile Plot: Using the 3D volume reconstructions for each MRI sequence as a reference (figures 3 and 4), the reformatted transverse reslice through the mid-height of each marker was located (figure 4B). A line selection (figure 4C) was drawn through the marker, such that the mid-point on the line was positioned at the interface between the marker and the skin and the endpoints were located in air and muscle tissue, respectively. The intensity profile along this line selection (figure 4D) was used to objectively compare the marker signal and visibility. The peak intensity in the marker, minimum intensity at the interface between the marker and tissue, mean intensity in the fat tissue and mean intensity in the muscle tissue were recorded from the profile.
3. Maximum Intensity z-projection: A two-dimensional (2D) image with each pixel intensity equivalent to the maximum intensity along a z-trajectory over a given range of stack images was created (figure 4E) The image range was defined to encompass the upper and lower limits of the marker, as viewed on the 3D volume reconstruction (figure 4A, orange broken line).
4. Intensity Histogram and Maximum Marker Intensity: A segmented line tool (figure 4F) was used to demarcate

| Table 2 | MRI sequence details | | | | |
|---|---|---|---|---|---|
| | Imaging protocol | Acquisition plane | Field of view (mm) | Repetition time/echo time (msec) | Resolution (pixel width x height x slice spacing, mm) |
| 1 | Proton density | Coronal | 300×300×44 | 2435/35 | 0.1465×0.1465×2.2 |
| 2 | T1-weighted, Spin echo | Coronal | 300×300×44 | 661/10 | 0.1465×0.1465×2.2 |
| 3 | T1-weighted, 3D fast field echo | Sagittal | 300×300×175 | 25/1.4 | 0.469×0.469×0.5 |
| 4 | T2-weighted, fat suppressed | Coronal | 300×300×44 | 4448/65 | 0.195×0.195×2.2 |
| 5 | T2-weighted, 3D | Sagittal | 300×300×175 | 2473/72 | 0.469×0.469×0.5 |

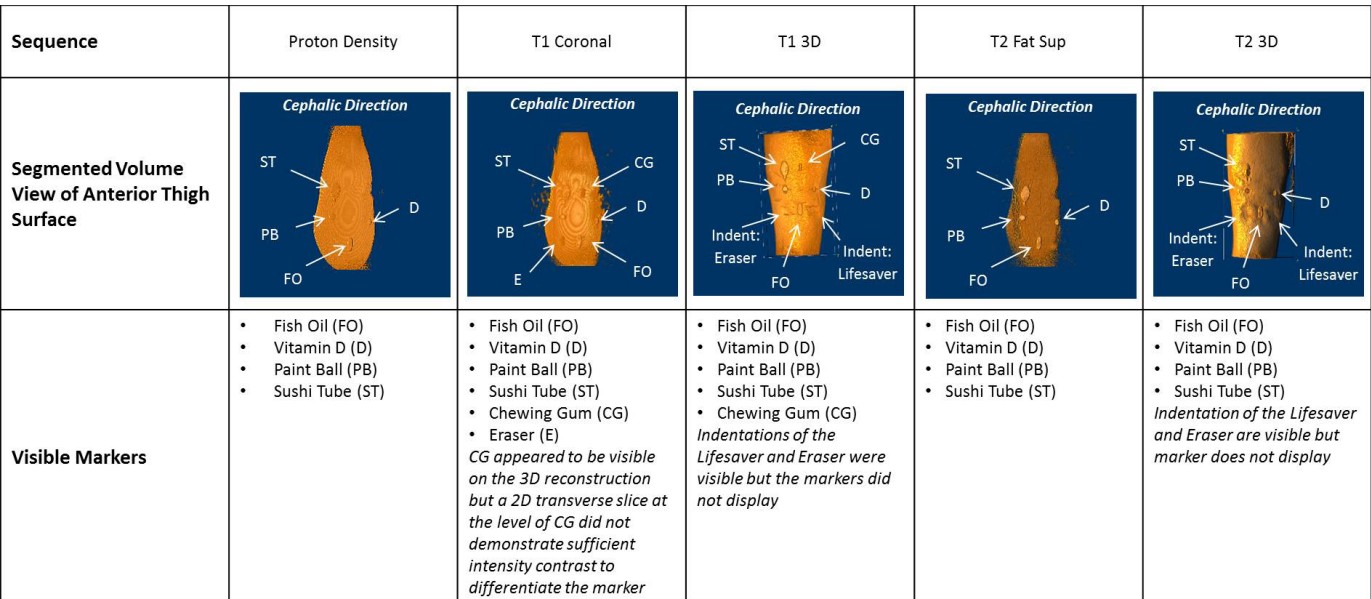

| Sequence | Proton Density | T1 Coronal | T1 3D | T2 Fat Sup | T2 3D |
|---|---|---|---|---|---|
| **Segmented Volume View of Anterior Thigh Surface** | *Cephalic Direction* ST, PB, FO, D | *Cephalic Direction* ST, PB, FO, E, CG, D | *Cephalic Direction* ST, PB, Indent: Eraser, CG, D, Indent: Lifesaver, FO | *Cephalic Direction* ST, PB, FO, D | *Cephalic Direction* ST, PB, Indent: Eraser, D, Indent: Lifesaver, FO |
| **Visible Markers** | • Fish Oil (FO)<br>• Vitamin D (D)<br>• Paint Ball (PB)<br>• Sushi Tube (ST) | • Fish Oil (FO)<br>• Vitamin D (D)<br>• Paint Ball (PB)<br>• Sushi Tube (ST)<br>• Chewing Gum (CG)<br>• Eraser (E)<br>*CG appeared to be visible on the 3D reconstruction but a 2D transverse slice at the level of CG did not demonstrate sufficient intensity contrast to differentiate the marker* | • Fish Oil (FO)<br>• Vitamin D (D)<br>• Paint Ball (PB)<br>• Sushi Tube (ST)<br>• Chewing Gum (CG)<br>*Indentations of the Lifesaver and Eraser were visible but the markers did not display* | • Fish Oil (FO)<br>• Vitamin D (D)<br>• Paint Ball (PB)<br>• Sushi Tube (ST) | • Fish Oil (FO)<br>• Vitamin D (D)<br>• Paint Ball (PB)<br>• Sushi Tube (ST)<br>*Indentation of the Lifesaver and Eraser are visible but marker does not display* |

**Figure 3** Segmented volume reconstructions of the anterior thigh surface, showing an example of markers that were visible for each MRI sequence for the first acquisition of 9 markers as shown in figure 2.

the outer profile of the marker on the 2D z-projection image (figure 4E). A histogram showing the distribution of maximum intensities within this profile for the marker was used to record the maximum and mean marker intensity (figure 4G).

In some instances (figure 3: T1_3D—chewing gum; T1 coronal—eraser), the intensity range exhibited by the marker was very low in comparison to the background intensity, making the marker boundaries difficult to differentiate from the surrounding regions. However, these markers were still included in the analysis as it was

possible to create a profile plot (figure 4D) on a transverse plane through the mid-height of the marker, with a measurable intensity gradient across the marker.

## Assessment of intensity values

To objectively assess whether the markers provided an acceptable visualisation on the images produced for each MRI sequence analysed, three criteria were defined to assess the results. It was expected that the markers should (a) provide a pixel intensity that was sufficiently above the background intensity of air to

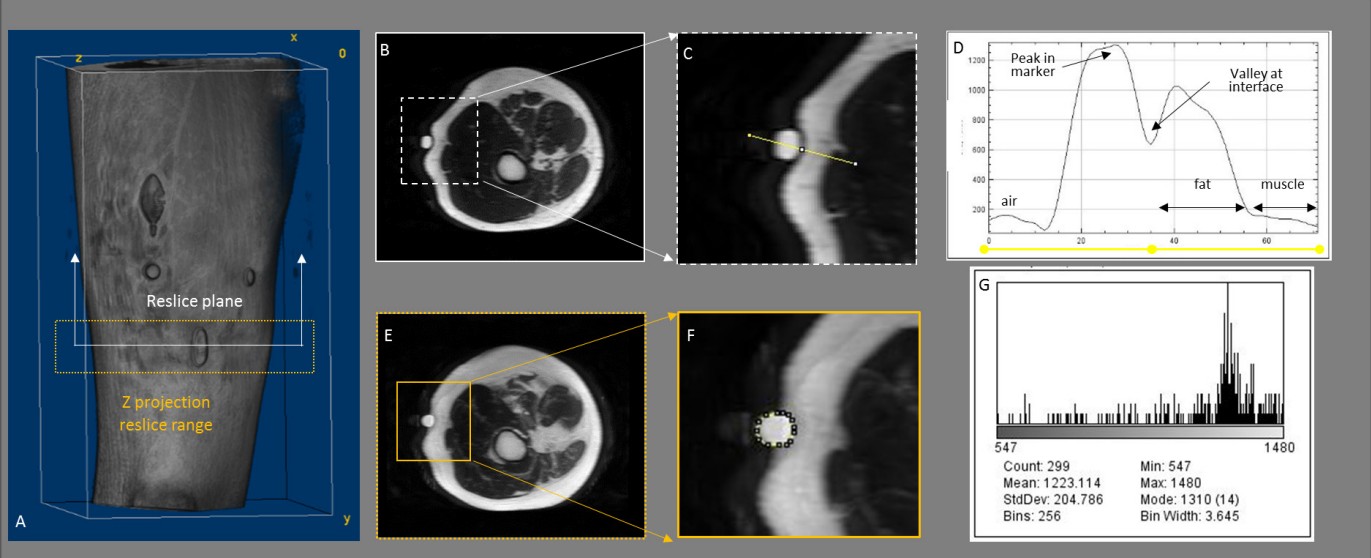

**Figure 4** Workflow to measure relevant intensity parameters from each MRI dataset for each visible marker. (A) 3D volume reconstruction of the T2, 3D sequence showing the transverse reslice plane (white line) and the range of z-projection reslice plane (orange rectangle); (B) transverse reslice plane through the fish oil marker; (C) line selection across the marker (yellow unbroken line); (D) profile plot of intensity values along the line selection; (E) maximum intensity, transverse plane z-projection of the fish oil capsule; (F) segmented selection tool demarcating the marker boundary (yellow broken line); (G) histogram detailing distribution of maximum intensities within the marker.

result in a clear visual contrast; (b) demonstrate a pixel intensity ratio at the interface between the marker and adjacent epidermis/fat tissue (ie, peak marker intensity divided by interface intensity/mean fat tissue intensity) greater than 1 and (c) demonstrate a pixel intensity ratio between the marker and the muscle tissue (ie, peak marker intensity divided by mean muscle intensity) greater than 1.

The ratios were intended to provide a numeric value that aligned with the perceptual and visual decision-making process a person undertakes when determining whether a marker is visualised at an acceptable quality. To indicate the quality of marker visualisation, this ratio of unity was defined after viewing all markers over the five MRI sequences and in combination with the visual assessment of visibility rating.

Depending on the MRI sequence, the fat and muscle tissue adjacent to the marker displayed over different intensity ranges. The ratio between the fat or muscle tissue and marker intensity was considered critical to be able to differentiate the marker from its surroundings. For this reason, the intensity ratio was considered of more importance than the absolute value of the pixel intensity.

## Patient and public involvement

This research was done without patient or public involvement as an aspect of a larger study yet to be performed. Participants in the larger study will be asked to be involved in the design of the subjective assessment methods.

## RESULTS

Of the 18 markers investigated, five were consistently visible for all imaging sequences, but this visibility was of differing quality (table 3). The FO, vitamin D (D), PB, soy sauce sushi tube (ST) and commercial marker (CM) were typically visible to a high quality (ie, rating 2 or 3) for all sequences (table 3). A visibility rating of 1 on only one of the MRI sequences was observed for the eraser (E), chewing gum (CG), lifesaver (LS) and coffee bean (CB), indicating the quality of visibility was poor. Surprisingly, the plasticine (PL) marker was visible to a high quality (rating 3) for the T1_3D sequence, but had little (rating=1 for proton density (PD) sequence) or no visibility for the other sequences. Similarly, the jelly baby (JB) was also remarkable in being well visualised on the T1_3D sequence with a rating of 2, but had no visibility scored for the other MRI sequences.

A mean visibility score greater than 2 was observed for the PB, FO, D and CM markers (table 3).

Figure 5 shows the summary of the visible markers on the five MRI sequences. In each case, the intensity range for the image was selected such that the intensity contrast within the image could best display the marker. These images do not include image filtering or contrast enhancement. Viewing the reslice images in figure 5 in combination with the visibility ratings in table 3 demonstrated that the best quality marker visibility for the T1_3D, T1 coronal, T2 Fat_Sup and proton density sequences was the PB, ST, FO and D; and for the T2_3D sequence was the PB, FO and D. The CM was best visualised on the T1_3D and T2 Fat_Sup sequences.

**Table 3** Visibility rating for each marker on each MRI sequence. If not visible, the marker rating=0. If visible, marker visibility rating was either 1 ('visible, but not easily'), 2 ('edges visible but fuzzy') or 3 ('very clear')

| | Proton density | T1 coronal | T1_3D | T2 Fat_Sup | T2_3D | Mean rating |
|---|---|---|---|---|---|---|
| Commercial marker | 2 | 2 | 3 | 3 | 2 | 2.4 |
| Fish oil | 2 | 2 | 3 | 2 | 3 | 2.4 |
| Vitamin D | 2 | 2 | 3 | 3 | 3 | 2.6 |
| Paint ball | 2 | 2 | 3 | 3 | 3 | 2.6 |
| Soy sauce sushi tube | 2 | 2 | 3 | 2 | 1 | 2.0 |
| Plasticine | 1 | 0 | 3 | 0 | 0 | 0.8 |
| Jelly baby | 0 | 0 | 2 | 0 | 0 | 0.4 |
| Coffee bean | 1 | 0 | 0 | 0 | 0 | 0.2 |
| Eraser | 0 | 1 | 0 | 0 | 0 | 0.2 |
| Chewing gum | 0 | 0 | 1 | 0 | 0 | 0.2 |
| Lifesaver | 0 | 0 | 0 | 1 | 0 | 0.2 |
| Fruit lolly | 0 | 0 | 0 | 0 | 0 | 0 |
| Blu-tak | 0 | 0 | 0 | 0 | 0 | 0 |
| Magnesium | 0 | 0 | 0 | 0 | 0 | 0 |
| Tic tac | 0 | 0 | 0 | 0 | 0 | 0 |
| Mentos | 0 | 0 | 0 | 0 | 0 | 0 |
| Mintie | 0 | 0 | 0 | 0 | 0 | 0 |
| Nail polish | 0 | 0 | 0 | 0 | 0 | 0 |

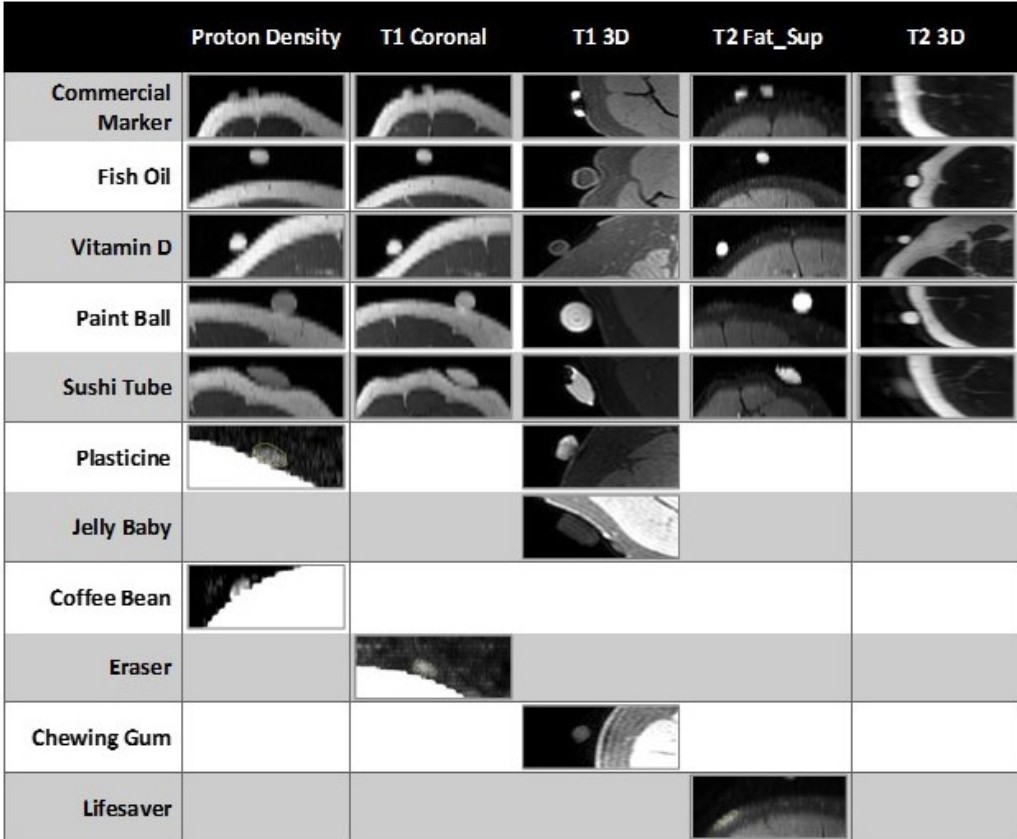

**Figure 5** Reformatted transverse reslice images through the mid-height of all visible markers on the MRI sequences analysed. Of the 18 markers, only 11 were visible on images from at least one of the MRI sequences—results for these 11 visible markers are shown. A blank box indicates that the marker tested was not clearly visible for a particular MRI sequence.

Intensity ratios between the maximum marker intensity and the mean intensity in either the fat tissue or muscle tissue are shown in figure 6.

For the PD sequence, while the CM, FO, D, PB and ST markers were easily visible (figures 5 and 6), the marker intensity was numerically similar to the fat tissue, resulting in marker-to-fat ratios near or below one. The PL and CB were less easily differentiated from the nearby fat tissue (figure 6).

For the T2 fat-suppressed sequence, there was a distinct differential between the maximum intensity in the marker and the mean intensity in the fat tissue, making all observable markers (CM, FO, D, PB, ST and LS) clearly visible (figure 6).

The CM was not clearly visualised for the 3D T2-weighted sequence (table 3, figure 5), and had a marker-to-fat ratio <1 (figure 6). While the ST tended to result in good visibility (table 3, figure 5), for the 3D T2-weighted sequence the marker intensity was less than the muscle tissue and not easily differentiated from the background air intensity. Aside from this, the 3D T2-weighted sequence showed an intensity ratio >>1 and thus clear intensity contrast for the marker-to-muscle ratio (>1) and marker-to-fat ratio, for all markers.

While the eraser (E) was visible on the T1 coronal sequence, the intensity ratios suggested that the intensity contrast between the marker and adjacent muscle/

fat was not of a good quality (table 3, figure 5). This was supported by the observed intensity ratio <1 for both the marker-to-muscle and marker-to-fat ratios (figure 6). For the other visible markers, the intensity ratios were all ≥1.

With the exception of the CG and JB, the intensity ratios for marker-to-fat were all >1, for the 3D_T1-weighted sequence. In the case of the PB and ST marker, since these were comprised a non-oil-based substance, the intensity contrast and marker-to-fat/marker-to-muscle intensity ratios were >>1 (figure 6). These latter two markers also resulted in a high visibility rating (table 3, figure 5).

## DISCUSSION

Fiducial markers are used for calibration of the imaging equipment, providing a reference measurement for templating software, locating specific boundaries for radiotherapy planning, orienting an image and very importantly, to identify a specific area of interest in the case of diagnostic queries. It is thus essential to have a marker that is detectable and clearly discernible from bone and soft tissues when placed in the visual field for a number of common MRI sequences. Single use, commercially available fiducial markers make up a significant component of the imaging department's outgoings and in an attempt to reduce costs, the current study was a practical exercise designed to image a number of commonly available

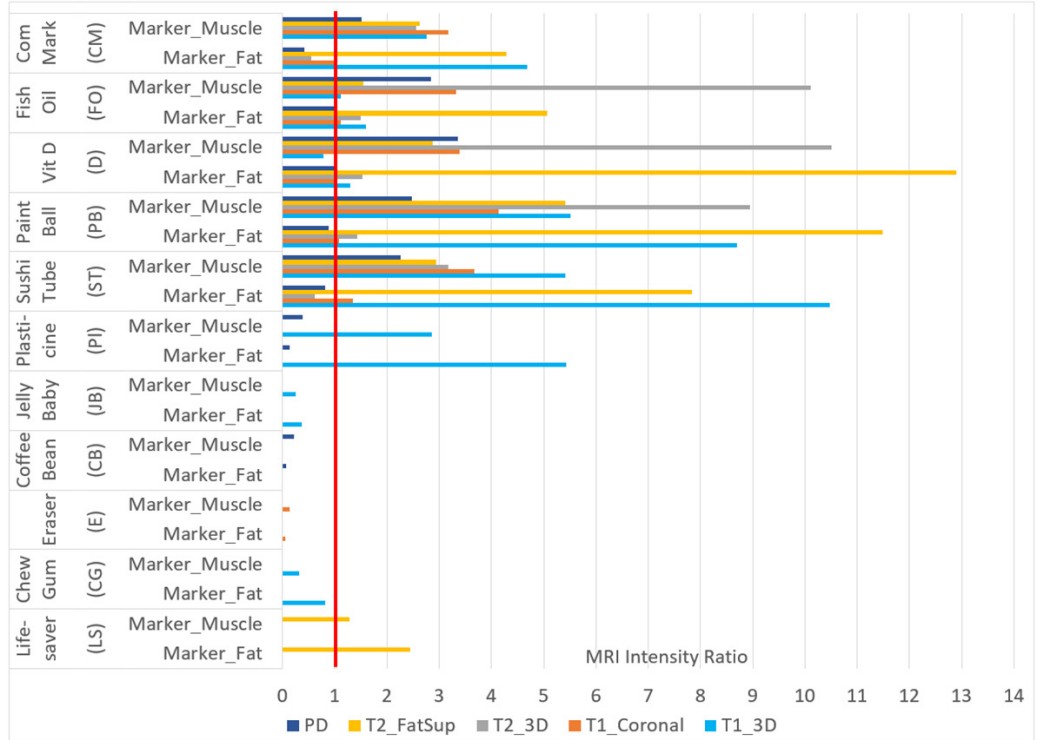

**Figure 6** Intensity ratios for all visible markers. Marker_Muscle and Marker_Fat are the ratios between maximum marker intensity and mean intensity in the fat tissue and muscle tissue, respectively. Results are shown for the five MRI sequences: PD, T2_FatSup, T2_3D, T1_coronal and T1_3D. PD, proton density; T2_FatSup, T2-weighted fat-suppressed; T2_3D, 3D T2-weighted; T1_coronal, T1-weighted coronal; T1_3D, 3D-T1-weighted.

everyday items to assess their suitability to function as an effective surface fiducial MRI marker substitute. The current study aimed to provide additional marker options useful for personalised musculoskeletal kinematic models as well as for clinical pathologies and research projects utilising MRI.

A single use, commercially available surface fiducial marker was evaluated in this study alongside 17 potential alternatives and its performance was surprisingly less impressive than some of the more common and affordable items tested. The CM is considered prohibitively expensive (AUD\$ 6–10 per marker) and is therefore used selectively for specific neurosurgical imaging needs only. Makeshift markers of various types have been anecdotally trialled in clinical radiology departments over the years. Discussions with our hospital imaging staff revealed that almonds, vitamin E capsules and condiment packets had previously been trialled. Packaging rupture was a risk and unfortunately these makeshift markers caused issues with patient comfort and inconsistent visibility on imaging.

In the current study, the FO, PB and D markers were all clearly visible on each of the five sequences and typically demonstrated well-defined intensity contrast with the adjacent fat/tissue. On the basis of intensity contrast, these three markers would be recommended in place of a CM for use as an external fiducial marker.

However, as noted in table 1, in addition to MRI visibility of the marker, consideration of the cost, accessibility and useability of the surrogate marker are of equal

importance. In light of this, the PB is less accessible, requiring specific online ordering or purchase at custom stores. Additionally, concerns remain with regards to the risk of damage to equipment and property if the PB ruptured as the pellets are designed to burst at low force to prevent injury during game play. The FO, while clearly visualised and sourced, are large in size making them difficult to use accurately for small pathologies or on extremities. They also bring with them the risk of capsule rupture and the release of a pungent odour.

Oil-based capsules are commonly used as a CM alternative.[20] Vitamin D capsules are economically purchased (cost of AUD\$36 for 400 capsules), small in size, readily available and bring no detrimental outcomes should they rupture. vitamin D provided a bright MRI marker virtually identical to the visibility of the FO capsule, but was significantly smaller in size (~13 mm long), making it more appealing when markers are required in imaging of the extremities, face or small pathologies. Anecdotally, the authors of this study have since used vitamin D capsules routinely for supine musculoskeletal studies of the spine requiring multiple markers affixed throughout extended sessions in the MRI scanner and have not had any degradation or rupture, with excellent visibility of these markers observed on the resulting MR images.

While the vitamin D capsule was the final fiducial marker of choice, a discussion of other markers that were well visualised on some sequences is relevant given the aim of this study. Fluids are more readily identifiable on

MRI and lipid-based markers have proven to be consistently reliable.[9 21] The capped sushi soy sauce tubes tested in the current study were reliably visible on all the MRI sequences analysed and were considered by the MRI radiographers as a useful alternative marker candidate. They suggested one or more of the fish-shaped tubes could act as a type of 'pointer' to highlight a small lesion or point of interest without obscuring it. It was also considered sufficiently small to not compress or cover the relevant anatomy.

Performing successful MRI on children can be challenging and may require a creative approach, especially in the younger age groups. Children who have experienced a lot of medical interventions can be fearful of any and all medical procedures. Therefore, we suggest it may be a great advantage to be able to produce a 'JB' sweet to apply as a fiducial imaging marker when a T1_3D sequence was required, or alternatively a fish-shaped soy sauce tube or vitamin D capsule if this particular sequence was not appropriate.

Depending on the reason for imaging and the specific pathology, different MRI sequences are chosen to provide well-visualised tissue contrast. For example, to demarcate the location of a tumour, a 3D T1-weighted/T2-weighted or T1 Fat-Suppressed protocol would be considered more appropriate while for exploratory imaging relating to tissue infection, a T1 coronal or T2 Fat-Suppressed protocol is preferable. Conversely, when MRI is requested for musculoskeletal conditions, a PD sequence is the preferred protocol to provide high intensity contrast in both fat and fluid in the bone. As such, this study provides evidence that there may not be one marker that is best suited to all imaging sequences and individual markers may provide good quality results for some but not all sequences (table 3, figure 5).

Regarding study limitations, while a single commercial fiducial marker was analysed in the current study, it was the brand used in our clinical imaging facility making it the most relevant marker to examine. A single human participant was used and only the thigh region was scanned, but we considered this to be superior to a saline phantom. Marker visibility was analysed by a single observer but typically in the clinical setting a single examiner views images to make judgements on visible anatomy. We acknowledge that some older clinical magnets may have a strength as low as 1.5T; however, we feel these findings relating to the FO, PB and D markers are still relevant for a lower strength magnet. Furthermore, there are a range of other additional MRI sequences that may be relevant for investigation of an alternative fiducial marker (eg, Dixon imaging); however, the five sequences chosen in the current study were on the advice of the collaborating Radiology Department and following a survey of local surgeons of the most commonly requested MRI sequences used for diagnostics and anatomical investigations.

**Contributors** MTI, CAG and JPL were involved in the conception, design and conduct of the research project, interpretation of the data and preparation of the manuscript. JPL performed the data analysis and edited the manuscript. DL wrote the first draft of the introduction and discussion. SM assisted with the design of the project, performed the imaging and assisted with the interpretation of the data and results. All authors were involved in the drafting and revising of the manuscript and gave final approval of the final version. All authors meet the ICMJE criteria for authorship. MTI and JPL are guarantors for the work and conduct of the study.

**Funding** The Queensland University of Technology, Brisbane, Australia provided the funding for the Article Processing Charges levied for publication of this manuscript.

**Competing interests** None declared.

**Patient consent for publication** Obtained.

**Ethics approval** Ethical approval was obtained from Queensland University of Technology Human Research Ethics Committee (approval # 1700000335).

**Provenance and peer review** Not commissioned; externally peer reviewed.

**Data sharing statement** Data may be made available via the author's Institutional Research Data Storage Service (RDSS) after the paper has been published. Through the RDSS, non-institutional researchers may be granted access to this data after making a request to author (Dr J P Little, j2.little@qut.edu.au) and owner of the RDSS entry.

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
