## [Reviewer comments · BMJ Open]

ARTICLE DETAILS

TITLE (PROVISIONAL)	Determining a Reliably Visible and Inexpensive Surface Fiducial Marker for Use in Magnetic Resonance Imaging: a research study in a busy Australian Radiology Department.
AUTHORS	Izatt, Maree; Lees, Deborah; Mills, Susan; Grant, Caroline; Little, Judith

VERSION 1 - REVIEW

REVIEWER	OSAMU TANAKA Asahi University Hospital
REVIEW RETURNED	23-Oct-2018

GENERAL COMMENTS	You should see reports of Tanaka et al.
---

REVIEWER	G Cohen USA
REVIEW RETURNED	03-Mar-2019

GENERAL COMMENTS	It seems that visibility is affected by the size of the marker as well as its content. for example ST or JB perform well but are also multiple times larger than some of the other markers discussed. I don't think this would be practical for RT applications. On the other end - the nail polish seems like a very thin dry application, and would be very surprised if you did get a signal for it. After initial tests, most markers can be ruled out. what is the rationale for analyzing them all? pg 5 ln 11: metallic objects are MR conditional, not compatible. table 1 and text: what is sufficiently small? Please be quantitative as this has implications to accuracy in radiotherapy treatment targeting table 1: Ease of handling and fixation seem redundant page 7 ln 15: I believe most magnets in clinical use are 1.5T not 3T page 7 ln 35: remove line spacing
--

	page 12 In 16 and elsewhere: what do you mean by distortion of 'pathology'? figures 1 and 2 do not contribute much to the manuscript, and should be removed. The reader can see what was done from figure 3 alone. figures 4,5,6 and 7 are very small and hard to read. recommend converting figure 7 to a table figure 6 blank boxes should be replaced with relevant MR images Finally: a comment to the editorial office: It is unacceptable that the personal details of the volunteer are disclosed and her privacy thus violated.
--	--

REVIEWER	Timothy JP Bray University College London
REVIEW RETURNED	26-Mar-2019

GENERAL COMMENTS	This study aims to identify a cheap and consistently visible surface marker for MRI. The question is of practical relevance, and it seems that there are few existing studies addressing this issue. The main result of the study (that the fish oil capsule is highly visible) is useful. However, the manuscript is much longer than it needs to be. Furthermore some of the choices of markers seem rather outlandish - e.g. it is hard to imagine using a wrapped sweet or sushi capsule as a medical device. Furthermore, some of the markers do not seem to be motivated by any knowledge of the underlying physics (eg tictac, magnesium). A much more comprehensive and useful study could be performed quite easily by (1) focusing on markers which can realistically be used as medical devices and (2) using a wider range of sequences. On the latter point, it would be interesting to know how the different markers would behave with fat suppression, Dixon imaging, variations in T1 and T2 weighting, etc. Given that only one subject has been scanned, I imagine that the experiment could be repeated fairly easily. Specific comments follow: Abstract  - The 'main outcome measures' section needs to be improved - I think the intensity ratio or the qualitative visibility scores are probably the main outcome measures here? Introduction  - The problem is clearly stated but this section is rather long and could be shorter and more focused. - Is there a publicly-available description of the content of the existing commercial products? If so, it would be useful to state this as this would guide the choice of markers. - The section on wood, porcelain etc is unnecessarily detailed and could be shortened (it is rather unsurprising that these substances would be poor MRI markers given the lack of water/fat content). - The final paragraph in the introduction (on the physics of MRI) seems most relevant - it is clear that finding markers with fat or water in would be most useful!
---

	Methods  - Again this could be more brief - details such as 'Following the advice from senior MR radiographers' could be removed. - It is a good idea to include the quantitative assessments in addition to the qualitative grading. However, the description of the metrics used is too long and somewhat unclear. It would be simple and clear to use an accepted metric such as contrast-to-noise ratio (the authors have used peak marker intensity rather than mean). I don't think the signal intensity itself (as in Figure 7) is of value because this is highly dependent on the sequence and specific parameters used - CNR is much easier to interpret. - The choice of some of these items - e.g. mentos, eraser - is rather bizarre as they do not contain water or fat. Results  - The images are of poor quality and cannot easily be read. Discussion  - Again this could be shortened. - It would have been useful to include a wider range of sequences - this should be at least mentioned as a limitation. For example, it would have been interesting to see the properties of the fish oil on Dixon imaging.
--	---

VERSION 1 – AUTHOR RESPONSE

Reviewer #1:

See reports of Tanaka et al.

A relevant reference to Tanaka et al. (2017) has been included in the first paragraph in the Introduction.

Reviewer #2:

Visibility is affected by the size of the marker as well as its content. For example ST or JB perform well but are also multiple times larger than some of the other markers discussed. I don't think this would be practical for RT applications. On the other end - the nail polish seems like a very thin dry application, and would be very surprised if you did get a signal for it.

The alternative marker options tested, including ST, JB and Nail polish, was in close collaboration with our radiology department and radiographer co-author. The JB is approximately the same size as the commercial fiducial marker. Regarding the ST, while the tube itself has a major axis length approximately twice the diameter of a commercial marker, as a consequence of this study, our radiographers have continued to use the ST for specific applications where they require anatomical points of interest to be highlighted with a "pointer" (refer Discussion section, page 12, paragraph 1). Additionally, the focus for this paper was to provide a broad range of items that could be viably used as a fiducial marker for an MRI-based radiological study, not to replace the commercial fiducial marker in situations that require pinpoint accuracy or those used for targeted radiotherapy. To this end, surface fiducial markers are frequently used to simply highlight anatomical landmarks for musculoskeletal studies and in this instance, the ST marker would be a suitable size and shape with the added advantage of a 'pointer-style' shape. The JB would be a suitable marker to use with paediatric patients with the appropriate MR image sequence as indicated by our Figures - i.e. a T1 3D sequence.

After initial tests, most markers can be ruled out. What is the rationale for analyzing them all?

We thank the reviewer for this comment as it highlights the conceptual planning the authors of this paper underwent in deciding upon the relevant results for presentation. In the initial planning of this study, the authors sought input from multiple sources on what potential everyday items could serve as

a viable alternative to commercial surface fiducial markers. Our sources were both literature-based, from the medical physics discipline, MR radiologists as well as anecdotal. This being the case, multiple items were tested and the final items included in this paper were those that were most relevant on the basis of our prior anecdotal discussions with radiographers/surgeons and on the basis of the limited literature we could find on the topic of low cost fiducial marker alternatives.

- pg 5 line 11: metallic objects are MR conditional, not compatible:
Thanks for this clarification, the term has been changed in the manuscript.

- table 1 and text: what is sufficiently small? Please be quantitative as this has implications to accuracy in radiotherapy treatment targeting:
We agree that this is a subjective criteria and very dependent upon the particular anatomy or pathology of interest. The phrase "Sufficiently small not to obscure relevant anatomy" has been removed from Table 1.

- table 1: Ease of handling and fixation seem redundant:
The ability to easily and reliably affix the marker to the participant was essential, in order to permit registration of MR images with other imaging and assessment modalities for our research projects. As such we have decided this criteria should remain in the table. However, we acknowledge that Geometry is perhaps an unnecessary criteria overall and has been removed from the table. Table 1 has been updated accordingly.

- page 7 line 15: I believe most magnets in clinical use are 1.5T not 3T:
As noted on page 7, first paragraph (Methods), currently in Australia, the commercial and clinical magnets are typically 3T and new magnet purchases for both private and public hospitals are at least of this strength. To the best of our knowledge, there are only two 1.5 T magnets remaining in our large metropolitan city. To this end, we respectfully disagree and therefore believe the results of this study are contemporary and relevant both for current facilities and moving forward, for future capabilities. In response to the reviewers comment, we have included an additional statement in the Discussion (second last paragraph): We acknowledge that some clinical magnets may have a strength as low as 1.5T, however, we feel these findings relating to the FO, PB and D markers are still relevant for a lower strength magnet.

- page 7 line 35: remove line spacing:
Spacing has been removed.

- page 12 line 16: what do you mean by distortion of 'pathology'?:
This phrase, "distortion of pathology" is misleading as it was based on anecdotal information from our radiographer (co-author) and has been removed so the meaning of this sentence is more succinct (now page 11, 3rd paragraph)

- figures 1 and 2 do not contribute much to the manuscript, and could be removed. The reader can see what was done from figure 3 alone:
We appreciate the reviewer's thoughts and have removed Figure 2A as we agree this is superfluous given the presence of figure 2B. We feel that it is still important to keep one of the images which shows the project method, the orientation and positioning of the markers and a perspective of the size ratio against the volunteer's anatomy, so Figure 2B has been preserved but renamed as Figure 2. Figure 1 is the only complete listing of the objects analysed so needs to remain.

- figures 4,5,6 and 7 are very small and hard to read. Recommend converting figure 7 to a table:
We have improved the resolution of the figures. Figure 7 has been removed altogether, See reviewer #3 comments) who found the bar chart (now Figure 6) to be the most visually clear. We acknowledge

that the text could be larger so this has been actioned in the updated Figures 4,5, and 6, and note that in the electronic version, there should not be a problem for the reader.

- figure 6 blank boxes should be replaced with relevant MR images:

The blank boxes in this figure (now Figure 5) were deliberately included in the Table because it instantly indicates to the reader that these markers were not clearly visible for a particular object using a particular sequence. The authors wanted this to be very obvious from the first glance at the table. This table will be a useful lookup for radiographers seeking alternative marker options for these MR sequences. E.g. – which sequence will show up the Jelly Baby marker?

For clarity, an additional sentence has been added to the Figure caption for this figure (now Figure 5). “A blank box indicates that the marker tested was not clearly visible for a particular MR sequence“.

Reviewer #3:

In the first instance, we are grateful for the very detailed comments provided by Reviewer #3.

- The main result of the study (that the fish oil capsule is highly visible) is useful:

In summarising the study findings, we respectfully correct this reviewer as it was not the ‘fish oil capsule’ that was highlighted as the most useful and versatile marker from this study, rather the Vitamin D capsule, which is an extremely relevant distinction, as both these capsules show a clear and high intensity image on MRI, but the latter does not bring with it risk of malodorous rupture and is of a much smaller size.

- Furthermore some of the choices of markers seem rather outlandish - e.g. it is hard to imagine using a wrapped sweet or sushi capsule as a medical device. Furthermore, some of the markers do not seem to be motivated by any knowledge of the underlying physics (eg tictac, magnesium):

The choice of markers was specifically intended to provide an unusual and broad range of everyday items, that could be cheaply and easily sourced by radiographers and would provide for both paediatric and adult patients, a level of ‘whimsy’ to assist radiographers to put young and at times very unwell patients, at ease.

Our choice of items was not without intellectual thought and was specifically chosen to include items with either a higher liquid or oil content OR to address anecdotal suggestions from both radiographers and surgeons alike.

All authors on this study have had considerable experience in working with a busy public radiology department, particularly one where paediatric patients are regularly scanned and where budget issues can be a constant challenge. Without this prior exposure, it may be that the importance and impact of the current study may have been lost, but for those working clinically day to day in this area, it is tremendously valuable to be provided with an alternative to the expensive, single use fiducial markers. It will be especially useful in low socio-economic countries with minimal finances.

- A much more comprehensive and useful study could be performed quite easily by focusing on markers which can realistically be used as medical devices:

With regards the comment that a ‘much more comprehensive and useful study’ could be performed, we would argue that the aim of this project was to investigate a broad range of items readily available in a hospital precinct (eg. local chemist, local grocery store). Given that following the outcomes of our study, the Public Radiology department with whom we are working as well as other clinical research groups, have chosen to adopt the use of Vitamin D capsules and the soy sauce sushi tubes for their MRI studies as well as in some clinical imaging, we would argue the items are most certainly realistic and useful as medical devices in this context.

- On the latter point, it would be interesting to know how the different markers would behave with fat suppression, Dixon imaging, variations in T1 and T2 weighting, etc:

We wish to highlight that one of the sequences chosen to analyse included fat suppression (see Table

2 & new Figures 3 & 5) and the choice of T1/T2 weighting was upon the advice of our radiology collaborators and based on a survey of local surgeons/practices in terms of the most frequently requested MRI sequence for musculoskeletal and pathology-based MRI. While there are of course multiple variations to these sequences which could be investigated, the current study was not intended to provide radiographers with a modified sequence for imaging of specific anatomy, but rather to provide them with a fiducial marker that could be readily visualised in their existing and commonly used standard sequences.

The five sequences chosen are those most relevant for a broad range of anatomical imaging and were initially advised by our radiology department, who regularly sees a broad range of pathologies (eg. oncology, musculoskeletal, internal medicine).

- Abstract - The 'main outcome measures' section needs to be improved - I think the intensity ratio or the qualitative visibility scores are probably the main outcome measures here?

The structured abstract has been modified as suggested: "Outcome measures were based on quantitative assessment of a clear intensity contrast ratio between the marker and the adjacent tissue and a qualitative assessment of visibility via a 3-point scale."

- Introduction - The problem is clearly stated but this section is rather long and could be more focused:

The Introduction has been reduced and removed/modified text is both highlighted in grey or cross-through to indicate any change that has been actioned.

- Is there a publicly-available description of the content of the existing commercial products? If so, it would be useful to state this as this would guide the choice of markers.

No – the authors attempted to uncover details of the contents of the commercial markers, both via search engines, patent searches and through direct conversations with the manufacturers. In all instances, we were unsuccessful due to manufacturer IP.

- The section on wood, porcelain etc is unnecessarily detailed and could be shorted (it is rather unsurprising that these substances would be poor MRI markers given the lack of water/fat content). The section on wood, porcelain etc. is relevant, as when searching for literature on what types of materials tended to show a visible and definable intensity on MRI, these articles were typically focussed on patient diagnosis for the presence of foreign objects. Such objects noted to have been visible on MRI in the literature were typically wood, glass, etc. so these items were included in the current study. However, in response to this reviewer's comment, this section has been reduced in length and removed/modified text is both highlighted in grey or cross-through to indicate its removal.

- The final paragraph in the introduction (on the physics of MRI) seems most relevant - it is clear that finding markers with fat or water in would be most useful!

We thank the reviewer for noting this and indeed, the final paragraph was certainly a main focus for the study, but we deliberately wanted to include the broad range of items that we were informed may be a possible alternate surface fiducial markers, on the basis of both anecdotal and literature based investigations. This was the reason for the broad and unusual range of items included in the study.

- Methods - Again this could be more brief - details such as 'Following the advice from senior MR radiographers' could be removed.

Text in this paragraph (Page 7, paragraph 2) has been removed to make this description briefer.

- It is a good idea to include the quantitative assessments in addition to the qualitative grading. However, the description of the metrics used is too long and somewhat unclear. It would be simple and clear to use an accepted metric such as contrast-to-noise ratio (the authors have used peak marker intensity rather than mean). I don't think the signal intensity itself (as in Figure 7) is of value

because this is highly dependent on the sequence and specific parameters used - CNR is much easier to interpret.

We see the reviewer's point regarding peak signal intensity and this was included to provide a perspective on how the different markers compared. Indeed at the end of the Methods section, in subsection 'Assessment of Intensity Values' the authors had noted that the intensity ratio was of more importance than the absolute value of the pixel intensity and Figure 7 was included largely for reference. In response to the reviewers comment, we have removed Figure 7 and kept only the Intensity Ratio Figure 8 (now Figure 6).

In contemplating the descriptive parameters used to express the visibility of the markers, we considered use of SNR and CNR. It was preferable to use a parameter which included a comparative measure of the signal between the marker and the adjacent tissue of interest. While CNR provides a measure of clinical image contrast, there was a level of subjectivity associated with defining the areal region of interest demarcating the adjacent tissue – this is important when calculating the mean signal in the reference region (ie. either the muscle or adipose tissue). Tissue gradation and muscle boundaries in these regions resulted in variability in the mean intensity and resulted in differing CNR values. For this reason, it was preferable to use a lineal selection method, with clearly defined and repeatable methods for locating the line (as noted in Figure 4), to avoid this variability. The peak intensity ratio was well suited to the current study requirements, provided a similar indication of marker image contrast to CNR and results were comparable with the qualitative findings and with the visual results shown in Figure 5.

- The choice of some of these items - e.g. mentos, eraser - is rather bizarre as they do not contain water or fat.

In the case of the Mentos sweet, they have a low percentage of fluid content evenly distributed throughout the capsule (as opposed to concentrated towards the centre) and their size and shape was appealing. In the instance of the eraser, this was upon the anecdotal advice of the MR radiographers, who noted they had visualised erasers on MRI previously. As mentioned, our choice of items was in all instances related to the points noted in Table 1 and moreover, based on prior literature investigations, foreign body literature and anecdotal suggestions from radiographers.

- Results - The images are of poor quality and cannot easily be read.

We have improved the resolution of the figures. The original manuscript Figure 7 has been removed and the new Figure 6 (previously Figure 8) has been improved to better visualise results.

-Discussion - Again this could be shortened.

The discussion has been shortened and removed/modified text is both highlighted in grey or cross-through to indicate its removal.

- It would have been useful to include a wider range of sequences - this should be at least mentioned as a limitation. For example, it would have been interesting to see the properties of the fish oil on Dixon imaging:

The following has been included in the second last paragraph of the Discussion section: Furthermore, there are a range of other MRI sequences that may be relevant for investigation of an alternative fiducial marker (eg. Dixon imaging), however, the five sequences chosen in the current study were on the advice of the collaborating Radiology department and following a survey of local surgeons of the most commonly requested clinical MRI sequences used for diagnostics and anatomical investigations.

VERSION 2 – REVIEW

REVIEWER	Timothy JP Bray University College London
REVIEW RETURNED	04-May-2019

GENERAL COMMENTS	The manuscript has been significantly improved since the first draft. The objectives are now clearer and the outcome measures are more suited to the question. The figures are much improved and illustrate the findings clearly. There are some remaining areas where the structure/organisation of the manuscript needs improving prior to publication. As it stands the structure of the paper is rather unconventional. Specific points follow - - The points in the 'strengths and limitations' section do not all fit under this title - some of the points read more like methodology- There are some referencing errors (some of the references have been replaced with an error message)- The hypothesis should be stated explicitly at the end of the introduction- There is too much interpretation in the results - this should largely be a statement of the findings (in the most objective fashion possible). For example, the comment 'a mean visibility rating greater than 2 was preferable' should be stated in the methods rather than the results. There are several similar examples in this section.- Much of the discussion is a 'general' discussion of the issues rather than relating to the results of the study. A discussion of the findings of the work here should come higher up and feature more prominently. The last paragraph ('In the current study, the FO, PB and D markers were all clearly visible...') contains key information that should come much earlier in the discussion.- There should be a clearly defined limitations section (this may reflect, for example, some of the issues around object selection which the authors have commented on in their response to reviewers)
--

REVIEWER	Gil'ad N Cohen MSKCC, NY, NY, USA
REVIEW RETURNED	13-May-2019

GENERAL COMMENTS	The authors have addressed my prior concerns in the revised manuscript.
---

VERSION 2 – AUTHOR RESPONSE

Reviewer(s)' Comments to Author:

Reviewer: 3

Reviewer Name: Timothy JP Bray

Institution and Country: University College London Please state any competing interests or state

'None declared': None declared

Please leave your comments for the authors below:

The manuscript has been significantly improved since the first draft. The objectives are now clearer and the outcome measures are more suited to the question. The figures are much improved and illustrate the findings clearly. There are some remaining areas where the structure/organisation of the manuscript needs improving prior to publication. As it stands the structure of the paper is rather unconventional. Specific points follow -

- The points in the 'strengths and limitations' section do not all fit under this title - some of the points read more like methodology.

The study strengths and limitations have been updated as follows, with changes highlighted in grey and the inclusion of a limitation:

- This manuscript is the first to test for easily sourced and economical items to find reliably visible surface fiducial marker options for common MRI sequences
- Scanning the fiducial markers options on the thigh of a healthy adult female provides a superior assessment of marker visibility than analyses that use a saline phantom object.
- The study presented both quantitative and qualitative analyses of marker visibility, thus providing a more practical assessment of the ability of the radiographer/investigator to visualise the marker.
- All imaging was carried out on a 3T MRI scanner, which may be of a higher strength than is typically used in lower socio-economic tertiary care institutions.
- Five commonly utilised MRI sequences were investigated to provide the most relevant alternative marker for MR clinical imaging.

- There are some referencing errors (some of the references have been replaced with an error message)

Thank you for alerting us to some persistent hyperlinks to Table 3 in a paragraph of the Results that have now been corrected. There were no Referencing errors.

- The hypothesis should be stated explicitly at the end of the introduction.

The following has been included as the last sentence in the Introduction.

We hypothesised that an inexpensive, readily sourced, robust surface fiducial marker could be isolated, that consistently demonstrated at least the same level of visibility as a commercial fiducial marker, when viewed on the most commonly performed MRI sequences.

- There is too much interpretation in the results - this should largely be a statement of the findings (in the most objective fashion possible). For example, the comment 'a mean visibility rating greater than 2 was preferable' should be stated in the methods rather than the results. There are several similar examples in this section.

The following has been added into the Methods, on Page 7 and similar text has been removed from the Results:

Using this scale, a mean visibility rating across the five MRI sequences was calculated for each fiducial marker. While a mean visibility rating >0 indicated that the marker could be seen on at least

one of the MRI sequences, a mean visibility rating greater than 2 was preferable as it indicated a consistently good quality visibility on all of the MRI sequences performed.

- Much of the discussion is a 'general' discussion of the issues rather than relating to the results of the study. A discussion of the findings of the work here should come higher up and feature more prominently. The last paragraph ('In the current study, the FO, PB and D markers were all clearly visible....') contains key information that should come much earlier in the discussion.

The Discussion has been majorly restructured and the main findings now appear earlier in the section. All changes are highlighted in grey and deletions are identified with 'strike-through'. The discussion now includes a clear description of the study findings, with examination of other useful findings relating to particular markers that may be well suited to specific applications.

- There should be a clearly defined limitations section (this may reflect, for example, some of the issues around object selection which the authors have commented on in their response to reviewers)

The last paragraph in the discussion notes limitations of the study and text has been included to highlight this as a 'limitations paragraph'.

Reviewer: 2

Reviewer Name: Gil'ad N Cohen

Institution and Country: MSKCC, NY, NY, USA Please state any competing interests or state 'None declared': None declared

VERSION 3 – REVIEW

REVIEWER	Timothy JP Bray University College London
REVIEW RETURNED	01-Jun-2019
GENERAL COMMENTS	The authors have significantly improved the manuscript and I would now recommend it for publication.

VERSION 3 – AUTHOR RESPONSE

Reviewer(s)' Comments to Author:

Reviewer: 3

The manuscript has been significantly improved since the first draft. The objectives are now clearer and the outcome measures are more suited to the question. The figures are much improved and illustrate the findings clearly and I would now recommend it for publication.